# Radiative Thermal Effects in Large Scale Additive Manufacturing of Polymers: Numerical and Experimental Investigations

**DOI:** 10.3390/ma15031052

**Published:** 2022-01-29

**Authors:** Benoît Cosson, André Chateau Akué Asséko, Lukas Pelzer, Christian Hopmann

**Affiliations:** 1IMT Nord Europe, Institut Mines Télécom, Centre Materials and Processes, University Lille, F-59653 Villeneuve d’Ascq, France; andre.akue.asseko@imt-nord-europe.fr; 2Institute for Plastics Processing (IKV) in Industry and Craft, RWTH Aachen University, Seffenter Weg 201, 52074 Aachen, Germany; lukas.pelzer@ikv.rwth-aachen.de (L.P.); zentrale@ikv.rwth-aachen.de (C.H.)

**Keywords:** additive manufacturing, radiative transfer, large scale, IR measurement

## Abstract

The present paper addresses experimental and numerical investigations of a Large Scale Additive Manufacturing (LSAM) process using polymers. By producing large components without geometrical constraints quickly and economically, LSAM processes have the capability to revolutionize many industries. Accurate prediction and control of the thermal history is key for a successful manufacturing process and for achieving high quality and good mechanical properties of the manufactured part. During the LSAM process, the heat emitted by the nozzle leads to an increase in the temperature of the previously deposited layer, which prepares the surface for better adhesion of the new layer. It is therefore necessary to take into account this part of heat source in the transient heat transfer equation to correctly and completely describe the process and predict the temperature field of the manufactured part. The present study contributes to experimental investigations and numerical analysis during the LSAM process. During the process, two types of measurements are performed: firstly, the heat emitted by the nozzle is measured via a radiative heat sensor; secondly, the temperature field is measured using an infrared camera while varying the process speed. At the same time, a numerical simulation model is developed in order to validate the experimental results. The temperature fields of the manufactured parts computed by numerical simulations are in very good agreement with the temperature fields measured by infrared thermograph with the contribution of the nozzle’s heat exchange.

## 1. Introduction

The industrial use of Additive Manufacturing (AM) is growing rapidly in many areas. The technology’s applications range from medical to aerospace. AM allows the manufacturing of parts with complex shapes, difficult to achieve by conventional processes. However, especially for parts obtained by extrusion-based AM technologies such as Fused Deposition Modeling (FDM), mechanical properties of the manufactured parts are oftentimes inferior as compared to parts manufactured by conventional processes such as injection molding. To improve the mechanical properties of parts manufactured by FDM, two approaches can be used. The first one is to improve the intrinsic properties of the constitutive material of the filament by adding reinforcements of different kinds in the thermoplastic [1,2,3,4]. The second approach, especially when high mechanical properties are required, is the optimization of process parameters [5,6,7].

Among the process parameters, the temperature of the extrusion nozzle, as well as its speed of movement, and the temperature of the build plate should be adapted to the mechanical and rheological properties [8]. These parameters directly impact the quality of the manufactured part [9]. An increase of the environmental temperature or the addition of a local material heating system in the FDM process can also have an influence on the manufactured objects [10]. The nozzle temperature ($T_{N}$) is one of the parameters having the most effect on the quality and the resistance of the manufactured part [11], since the viscosity of the extruded material depends directly on it [9]. Indeed, by increasing this temperature during the AM process, several researchers [12,13] have observed a drop in viscosity, which then induces transverse flows of the material after deposition, thereby reducing roughness.

The surface quality is improved, as well as the strength of the bond between the different layers obtained by better diffusion through the interfaces. The thermal phenomena (conductive, convective and radiative exchanges) occurring during the process have also been studied by several authors [14,15]. They directly influence the surface quality, dimensional accuracy and mechanical properties of parts. These heat exchanges are among those to be taken into account for modeling and simulating the FDM process [14].

In other cases, the addition of a local laser heat source near the nozzle during extrusion deposition has also been studied by other researchers [16]. During manufacturing, the laser was positioned to heat the strands of the lower substrate layer just before depositing new material on top. The bond strength between layers was found to be increased by 50% and the mode of inter-layer failure became more ductile, with a visible presence of a plastic deformation zone [17].

During the AM process of large-scale parts, the nozzle also emits heat (radiation) on the previously deposited layer which increases in temperature and improves adhesion with the new layer. In one study [18], a desktop FDM machine was used to study the effect of nozzle radiation on the fused filament fabrication process. A proposed 3D numerical model provided information on how the nozzle radiation affects the temperature field of the manufactured part. The temperature fields of the parts computed by numerical simulations were in very good agreement with the temperature fields measured by infrared thermograph.

Based on the mentioned studies, it is evident that temperature conditions influence extrusion-based AM processes and the parts created by using said processes noticeably. To improve process understanding, choice of process parameters as well as predictability and reproducibility, with the goal of obtaining functional parts with high mechanical properties, it is necessary to regard all temperature-related aspects of the AM process. Therefore, this article combines experimental investigations with numerical simulations for a better understanding and improved control of thermal phenomena during extrusion-based AM processes.

## 2. Experimental Setup

The investigations were carried out using the hybrid manufacturing cell, developed by the institute for plastics processing, Aachen, Germany. It was created to address the major challenges additive manufacturing technologies have to face: production speed, limited part size and limited choice of materials. To improve on state-of-the-art machines, it consists of a screw-based extruder, used for plasticizing and extruding thermoplastic pellets, which is mounted to a six-axis robot arm, providing flexibility in movement and a large build volume. The machine is therefore capable of producing large components in a short production time [19]. The extruder consists of a three-zone screw, mounted inside a divided housing. The intake zone at the upper half of the extruder is water-cooled to prevent premature melting of the material. The lower half is heated by three 100 W heating cuffs which, together with the sheer forces applied by the screw, plasticize the thermoplastic material. The screw is driven by a servo motor coupled to a planetary gearbox. Because of the gear ratio of 320:1, sufficient torque can be applied. To shape the extruded plastic and apply it, layer by layer, to the part being produced, it is pushed through a nozzle with a defined bore. The nozzles used can either be conical or cylindrical in shape, depending on the application.

For example, for processing fiber-filled materials, a cylindrical nozzle geometry would be selected to prevent individual fibers from sticking out in build direction and therefore causing failed parts. On the other hand, if an unfilled material is being processed and the thermal stress on the component should be limited, a conical shape would be advantageous. In combination, the above-mentioned aspects allow for a high material throughput. It is capable of processing thermoplastic pellets, which are less expensive as compared to filament used in other machines. It therefore enables additive manufacturing using standard plastics used in other processes. Furthermore, high levels of filling materials, such as carbon fibers or glass fibers, are possible. Figure 1 shows the extruder and its components.

By using the six-axis robot arm, the extruder is moved over the build plate, which is 1900 mm by 600 mm in size. It consists of a 10 mm thick, precision milled aluminum plate which provides flatness across the whole surface. To allow good adhesion between part and build plate, it is covered with a 0.2 mm thin sheet of polypropylene. The sheet is white and semi-translucent. Optionally, the temperature of the joining zone can be influenced locally by either heating or cooling. This enables better layer adhesion as well as better dimensional accuracy of the manufactured component [10].

To measure heat radiation during manufacturing, boxes of 150 mm by 150 mm by 150 mm are produced using a nozzle with a 0.6 mm orifice. The wall thickness is set to be 1 mm and the layer height is configured as 0.3 mm. To assist bed adhesion, a brim of 10 perimeters is used. The boxes are manufactured without infill, bottom or top layers. The material used is the black polypropylene RA130E by manufacturer Borealis AG, Vienna, Austria. The target temperature of the three heaters are, from top to bottom, $265\;{\;}^{\circ }$C, $280\;{\;}^{\circ }$C and $335\;{\;}^{\circ }$C. The in plane displacement velocity of the nozzle is set to 100 mm·s${}^{-1}$.

Before manufacturing the boxes, calibration of the extruder is needed. This calibration step consist of measuring the emitted radiative heat flux of the extruder and nozzle, as shown in Figure 2a. For the measurement of this radiative heat flux, a radiative sensor by manufacturer Captec, Lille, France, is used. The effective sensor area is 100 mm by 100 mm with a sensitivity of 300 mV/(W/c m${}^{2}$). The data are recorded directly on a computer.

During this step, the extruder is heated but not filled with material. Therefore, no material is being extruded. The center of the sensor is located at the coordinate (0, 0, 0) and edges of the square sensor are parallel to *x* and *y* axes of the machine. The nozzle follows straight lines parallel to the *x* axis on 9 planes orthogonal to the *z* axis. In each plane, the nozzle follows 5 equidistant lines. The third line (in the middle) follows the equation y=0. At the middle of this line, the nozzle is directly above the center of the sensor. The aim of this measurement is to isolate and determine the proper radiative heat flux of the extruder during the AM process. In this configuration, the measurements are not perturbed by the heat source for the extruded thermoplastic part that is still in a high temperature state at the beginning of the cooling process. In order to determine the influence of the distance between the sensor and the extruder on the received radiative heat flux, a path of the extruder is designed. This path consist of several round-trips at several heights. A numerical simulation of this step is presented in the numerical modeling section and experimental and numerical data will be compared in the results section.

A complete monitoring of the AM process is proposed by using the radiative sensor and an IR camera. It is a IR camera (Flir CEDIP JADE) having a spectral response of 3.6 μm–5.1 μm, a sub windowing of 320 pixels × 256 pixels InSb focal plane array (FPA), a 50 mm fixed focus lens and a manufacturer rated precision of ±1%. A computer is connected to the camera for image grabbing and further data analysis with Altair software. This experiment is used to show the effect of self heating of AM parts and the effect of the extruder’s radiative heat flux on the temperature field evolution of the part during the process. The box described previously is used in this experiment. The investigations are focused on the temperature evolution of one of the four walls of the box (Figure 3). A simultaneous data recording of the heat flux and the temperature field is done.

## 3. Numerical Modeling

In order to explain and determine the radiative exchanges, observed and measured during the experimental developments, between the manufactured part and the extruder and between the manufactured part and the environment, a numerical model based on view factors is developed as in [20]. With this model and the data of material properties, nozzle geometry and temperature, a complete description of the LSAM process and the computation of the manufactured part’s temperature history is investigated.

For computing the different exchanges, the extruder, the manufactured part and the radiative sensor are discretized in planar surface elements. The surfaces of the extruder, the part and the sensor are discretized in planar face element. By this discretization, it is possible to add up the view factor of each element to the sensor (Equation (Equation 1)).

The real shape of the extruder (Figure 1) is very complex and, moreover, the emissivities of the different parts of the extruder are heterogeneous. In order to simplify the computation, the shape of the extruder is defined by a cylinder and a cone. The emissivity and the temperature field are kept constant and homogeneous (Figure 4). $O_{e}$ is the center of the extruder element, $O_{s}$ is the center of the sensor element and *O* is the coordinate system’s fixed center. In order to simulate the radiative heat transfer between the extruder and the sensor as well as between the part and the sensor, two numerical strategies are developed. The first case is corresponding to the determination of the radiative power emitted by the nozzle by measuring the heat exchange between the extruder and the sensor when the thermoplastic is not extruded. The geometrical configuration of the experiment, i.e., relative positions of sensor and extruder, is given in Figure 4 [21]. In this configuration, the nozzle path is defined as shown in Figure 5a. There is no obstacle between the extruder and the sensor. The scalar product between the normal vectors $\vec{N_{1}}$ and $\vec{N_{2}}$ of the two element has to be positive (Equation (Equation 2)).
(1)$$dF_{d1-d2}=\frac{\cos\left(\theta_{1}\right)\cos\left(\theta_{2}\right)}{\pi S^{2}}dA_{2}$$
(2)$$\langle \vec{N_{1}},\vec{N_{2}}\rangle >0$$

In the second case, the thermoplastic is extruded and a square box is manufactured as shown in Figure 3. The geometry of the box is given in Figure 5b. The sensor is placed in front of the manufactured part on the build platform. In the experimental setup, an IR camera records the temperature of the wall that is in the front of the sensor. This wall will have a radiative exchange with the sensor. The temperature of the wall can be in the range from room temperature (fully cooled) to extrusion temperature (newly extruded polymer). In this numerical development, the evolution of the wall’s height and temperature are taken into account. The wall’s temperature field is extracted from the IR record an then used to compute the radiative exchange between the box and the sensor. During the process, the height of the manufactured walls increase. When the height reaches a certain value, some parts of the extruder can be hidden and cannot be recorded by the sensor when the extruder follows its path around the square section of the manufactured box. For the view factor computation between two elements of the sensor and the extruder, the value is equal to zero when the element of the extruder is hidden by the wall. In order to compute the view factor value, the vector-based geometry is used. The view factor has to be computed for each couple of elements from the extruder and the sensor. If $n_{e}$ is the number of elements used to discretize the extruder and $n_{s}$ the number of elements used to discretize the sensor, $n_{e}\cdot n_{s}$ view factor computations have to be done to compute the total radiative power received by the sensor from the extruder at each time step. As shown on Figure 3, to know if the intersection between the segment ($\left[O_{e}O_{s}\right]$), formed by the centers of the two elements (extruder and sensor), and the plane of the wall is inside the rectangle formed by the vertex (named A, B, C, D). In order to compute the intersection between the segment $\left[O_{e}O_{s}\right]$ and the rectangle (ABCD), ABCD is divided in two triangles (ABC and CDA). The intersection is computed for ABC (respectively CDA) as follows: (3)$$\vec{V}=\frac{\vec{O_{e}O_{s}}}{∥\vec{O_{e}O_{s}}∥}\;;\;\vec{R}=\vec{OO_{e}}\;\vec{X}=[\vec{OB}-\vec{OA},\vec{OC}-\vec{OA},-\vec{V}]\backslash (\vec{R}-\vec{OA});$$



(4)
$$\begin{array}{c} \left\{\begin{array}{c} u=X(1)\\ v=X(2)\\ t=X(3) \end{array}\right. \end{array}\;Test=\left(u>0\right)\;\&\;\left(v>0\right)\;\&\;\left(u+v>1\right)\;\&\;\left(t>0\right)$$



The point of intersection between the segment and the plane is compared with the two triangles. If the point is inside one of the two triangles, the view factor value between the extruder and the sensor is equal to zero. The intersection between ABC respectively CDA and the segment $\left[O_{e}O_{s}\right]$ occurs when the value of Test is TRUE (Equation (Equation 4)).

In order to compare experimental data to numerical results, the computation of the power ($P_{n\to s}$ Equation (Equation 5)) received by the sensor from the nozzle is done for discretized locations on the nozzle path with a virtual displacement speed. The nozzle’s value of emissivity $\varepsilon_{n}$ and the part’s value of emissivity $\varepsilon_{w}$ are taken equal to 0.95 [18].
(5)$$\begin{array}{c} P_{n\to s}=\frac{1}{S_{s}}\int_{S_{s}}\int_{S_{n}}\varepsilon_{n}\sigma \left(T_{n}^{4}-T_{s}^{4}\right)dFdS\\ P_{w\to s}=\frac{1}{S_{s}}\int_{S_{s}}\int_{S_{w}}\varepsilon_{n}\sigma \left(T_{w}^{4}-T_{s}^{4}\right)dFdS\\ \sigma \approx 5.67\times 10^{-8}\;W\cdot m^{-2}\cdot K^{-4} \end{array}$$

The view factor computation depends only on the geometrical configuration in the study of the effect of the extruder on the sensor, the extruder is assumed to have a time independent temperature. This leads to a time independent emitted power. However, for the case of manufacturing the cubic box, where the part’s temperature decreases during the cooling process, the radiative power emitted by the box ($P_{w\to s}$ Equation (Equation 5)), that is a function of the temperature and the view factor, is changing with time. Moreover, the size of the part changes during the AM process (Figure 6b). In this case, the temperature field (recorded by IR camera) is heterogeneous and time dependent (Figure 6a).

## 4. Results and Discussion

The results of the calibration step where the radiative power emitted by the extruder is measured by the IR sensor is plotted in Figure 7. In this configuration, only the heaters are switched on. There is no extrusion of polymer through the nozzle. In both results, experimental and numerical, the effect of the relative position between the sensor and the nozzle is visible. Each local maximum of the curves corresponds to a position of the nozzle on (x=0) (Figure 5a). The global measured power decreases with the height of the nozzle (*Z* in Figure 5a). When the position is higher then 0.6 m, for a normalized time over 0.5, the radiative power received by the sensor from the extruder starts to be negligible (Figure 7). Normalized time describes the elapsed time since the start of the measurement divided by total time of the measurement. The view factor is proportional to the inverse square distance between the sensor and the extruder (Equation (Equation 1)). Therefore, for large parts, the effect of the nozzle on the evolution of its temperature field only has to be taking into account in a zone of interest that has a radius close to 0.6 m centered around the nozzle. Over 0.6 m the radiative effect of the nozzle is negligible. The results given by the view factor computation are in good agreement with the experimental results. In Figure 7, the local minimum of the received (or calculated) power, when the nozzle is not directly above the sensor, is minimal when the nozzle is in the plane at a height of 0.1 m (for a normalized time less than 0.15). This height is the closest height tested in this first study. Between the normalized time 0.15 and 0.6 the local minimum values of the received power increase. After 0.6, those values decrease. In a first reflection it can be suggested that the local minimum values decrease with the height of the nozzle. In fact, when the numerical results are observed, for small distances between the sensor an the nozzle, there is a competition between the product of the two cosines (the value of this product is equal to zero when Z=0), that is very small, and the square of the distance between the two elements nozzle and sensor (Equation (Equation 1)). This effect vanishes when the distance between the sensor and the nozzle has reached a certain value (here 0.8 m), and the values of the angles $\theta_{1}$ and $\theta_{2}$ are more stable and the growth of the term $S^{2}$ mainly influences the variation of the view factor value. For the following experiment, this observation is important to understand. It also shows that it is important to correctly compute the radiative emissions regarding the geometrical model used to describe the extruder and then correctly simulate the AM process [11].

Thermographic images where recorded as a video sequence with the infrared camera [22]. The temperature field recording of the manufactured part’s front face is done with an OPTRIS IR camera (Figure 6). This camera has a fixed focus lens. A post-treatment of the data is needed to create a continuous crop of the images in order to focus only on the already manufactured part and remove the nozzle and the background. In Figure 6a, the process is at 50% of the total manufacturing time. In Figure 6b, the thermography is taken at the end of the manufacturing process. These new data are synchronized with the nozzle position and the corresponding view factor computation between the nozzle and the IR sensor. In Figure 6a, the highest wall temperature is recorded at the same level of the extruder temperature [23]. This is the process temperature $T_{p}$. The lowest temperature is recorded on the build plate [24]. In Figure 6, the reflection of the radiative emission from the manufactured part can be seen on the build plate. The wall temperature is homogeneous in each horizontal plane. It continuously increases from the build plate to the last deposited layer. The temperature values range from room temperature to the temperature of the extruded polymer. Additionally, the manufactured part is submitted to heat exchanges by convection on every side and by conduction with the build plate.

While manufacturing the box, a brim of ten perimeters is deposited in the first layer, ensuring good adhesion between part build surface during the process (Figure 5b). During this step, the nozzle is close to the build plate and gets closer to the sensor over time (up to 0.05 in normalized time). In Figure 8a, as is not firstly expected, the power received by the sensor decrease while the minimum distance between the sensor and the extruder decrease. This effect is also visible in the numerical results (Figure 8b). This phenomenon is explained previously and it is related to the expression of the view factor (Equation 1) and the product of the cosines. In a second time, the value of the power measured by the sensor increase with the height of the manufactured box. The power value reaches its maximum around 0.3 (normalized time) and then decreases till the end of the process. It can be seen that the measured power has a minimum value when the extruder is manufacturing the wall placed at the opposite of the sensor.

In order to explain the different phases of the power evolution, the part due to the nozzle and the part due to the wall of the box are uncoupled and plotted on the Figure 9. In the first phase, the power is only received from the nozzle. There is only the bottom of the box bonded to the substrate. In the second phase, both of the power received from the nozzle and from the wall are increasing. In the third phase, the power due to the wall still increase to an asymptotic value and the power due to the nozzle decreases. In this phase, when the nozzle is at the opposite of the sensor, the minimum of the power due to the nozzle reaches zero. The height of the box reaches a point where it hides the nozzle from the sensor. The two components of the power, nozzle and wall, have to be simulated to understand the experimental results. The power received from the wall has a monotonic variation contrary to the one received from the nozzle that has an alternative variation [25].

## 5. Conclusions

In this study, developments have been done in numerical and experimental investigations on thermal exchange between the extruder and the printed part during large scale 3D printing. Several new features are presented:A numerical model coupled with experimental data was developed for the LSAM polymer process.Highlighting of radiative thermal exchanges that should be used for the simulation and optimization of extrusion-based AM processes.Confirmation of the importance of the infrared radiation emitted by the nozzle. This radiative energy can help to weld deposited strands on previously manufactured strands by increasing the interface temperature.By the numerical results, the radiative power emitted by the wall of the manufactured box is also highlighted.

As shown by the results given in this study, the self-heating during the production process of a part with a hollow shape should be taken into account for numerical simulation of the AM process. It is also shown that the radiative power emitted by the wall is not negligible. In future works, it should be necessary to add all this radiative thermal exchange in a numerical simulation of the material flow during the process in order to correctly compute the bonding between two adjacent strands.

## Figures and Tables

**Figure 1 materials-15-01052-f001:**
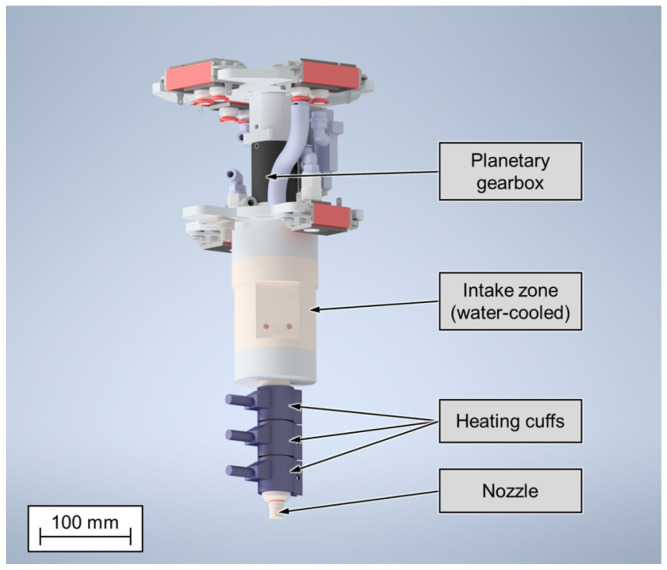
Screw-based extruder for processing thermoplastic pellets in additive manufacturing.

**Figure 2 materials-15-01052-f002:**
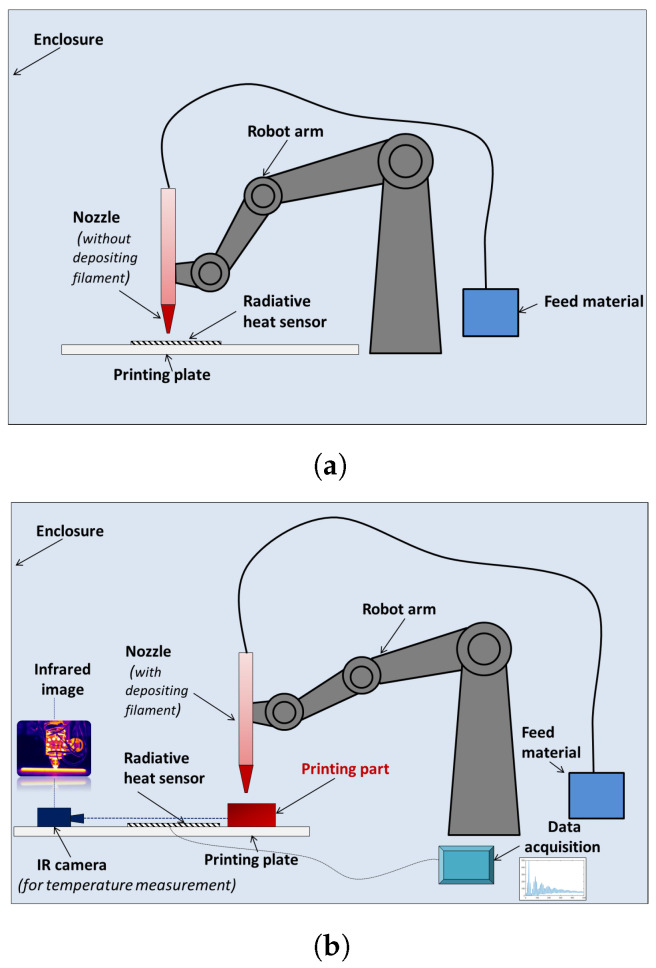
Experimental setup for determining the amount of thermal radiation during extrusion-based AM processes: (**a**) radiative calibration of the extruder with nozzle over the IR sensor without polymer extrusion, (**b**) infrared measurement from part and extruder during polymer extrusion.

**Figure 3 materials-15-01052-f003:**
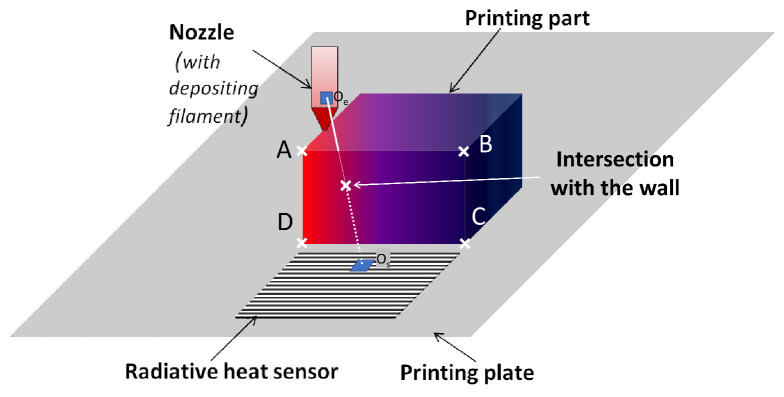
Radiative heat transfer measurement.

**Figure 4 materials-15-01052-f004:**
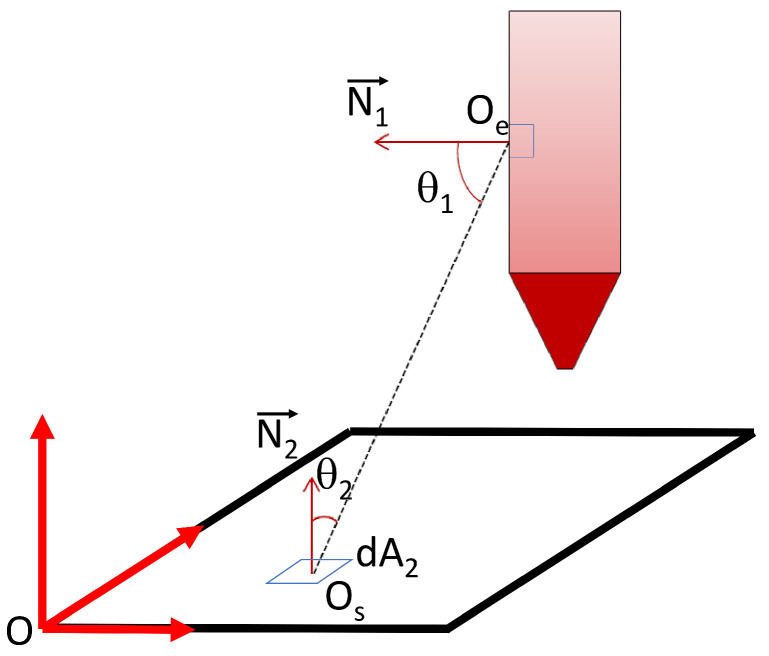
Geometrical configuration for the View factor computation.

**Figure 5 materials-15-01052-f005:**
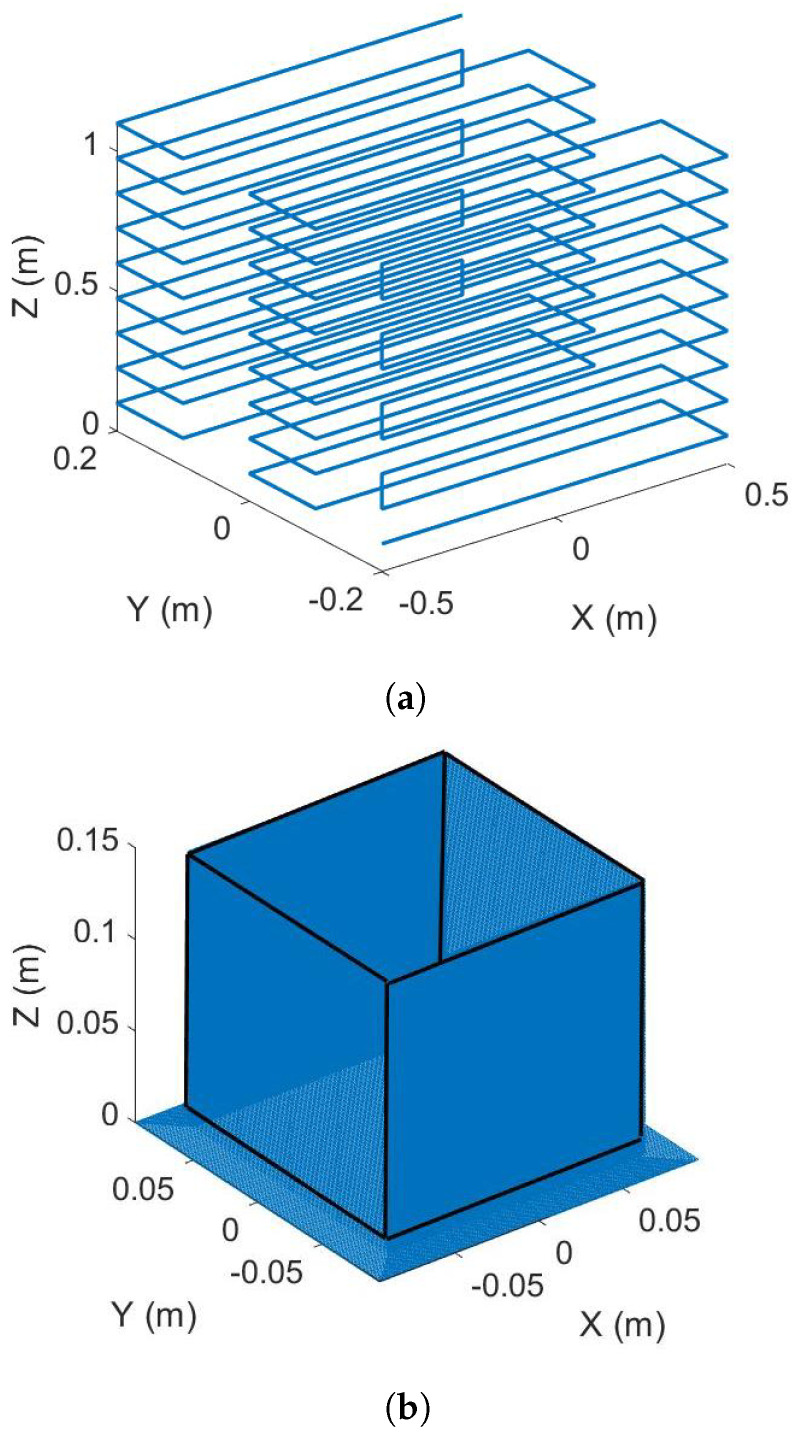
Numerical path of the extruder: (**a**) Path of the extruder over the sensor, (**b**) Path of the extruder to manufacture the part.

**Figure 6 materials-15-01052-f006:**
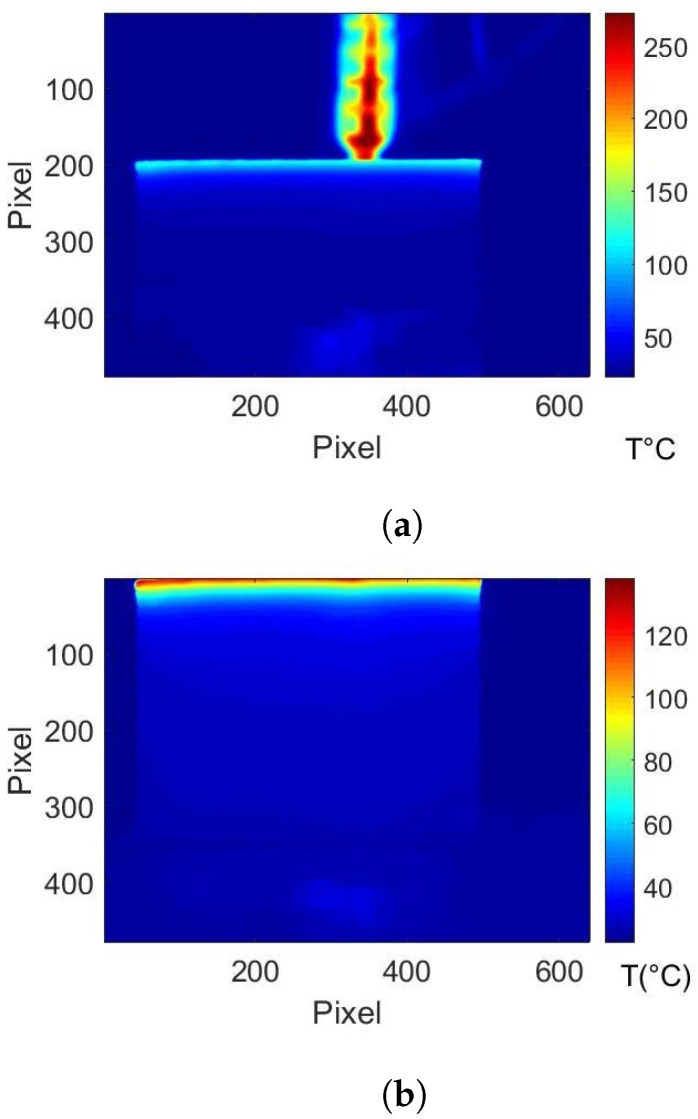
Thermography of the manufactured box: (**a**) at 50% of the process, (**b**) at the end of the process.

**Figure 7 materials-15-01052-f007:**
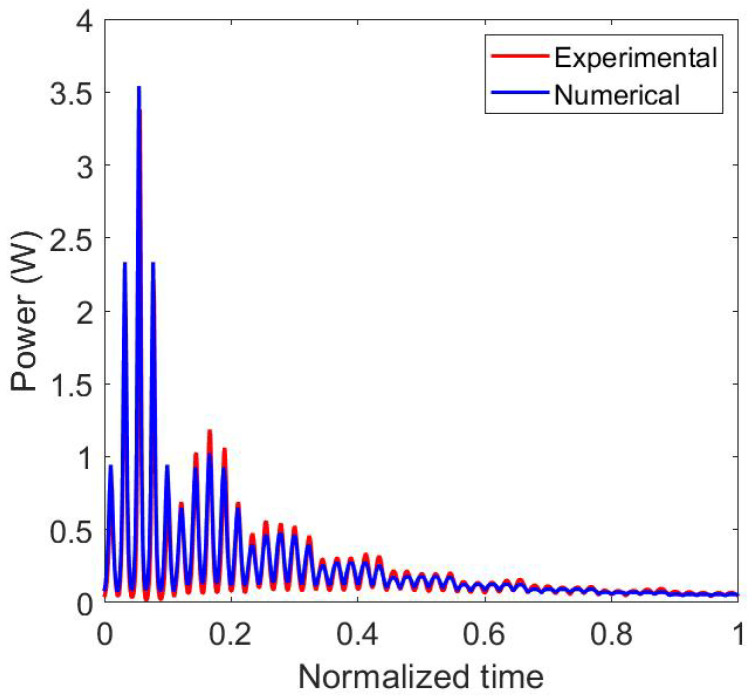
Comparison of experimental and numerical results for the nozzle calibration with the extruder without extrusion of polymer.

**Figure 8 materials-15-01052-f008:**
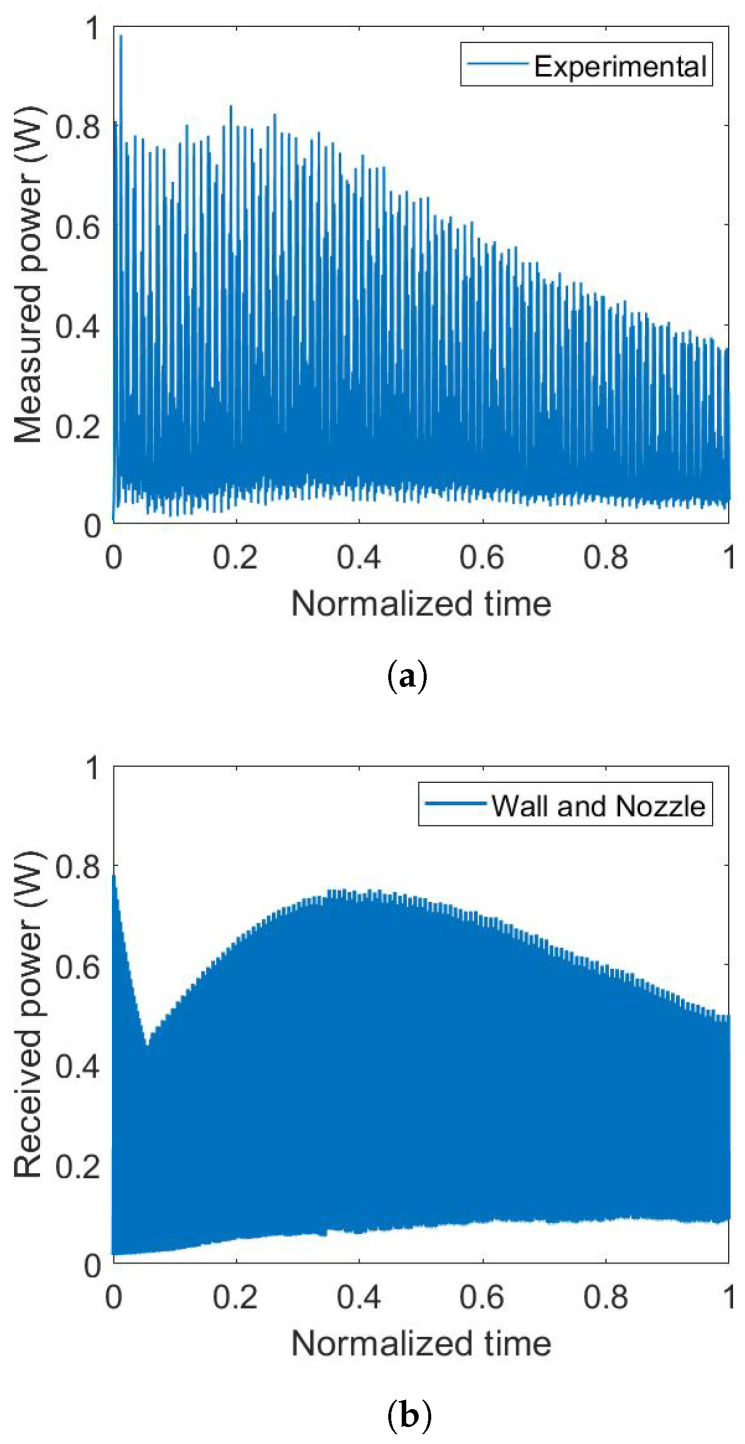
Radiative heat power on the sensor: (**a**) experimental results, (**b**) numerical results.

**Figure 9 materials-15-01052-f009:**
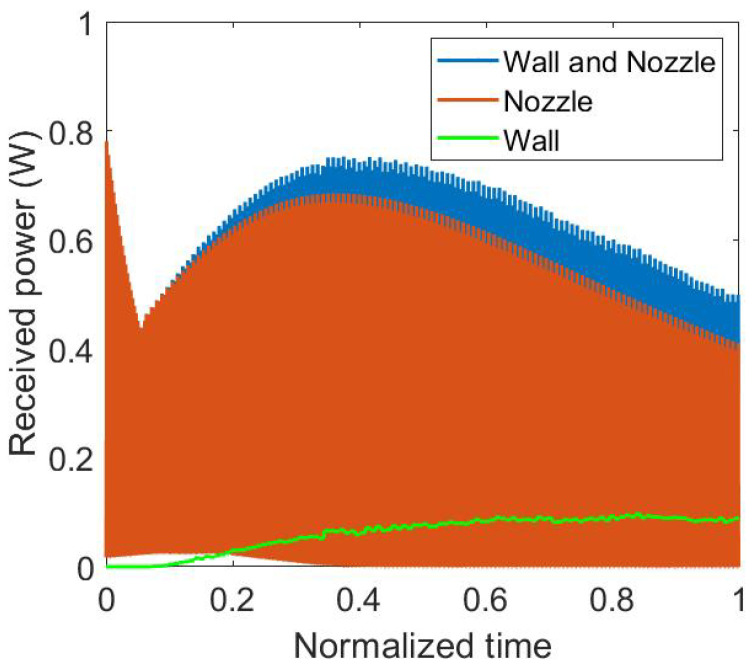
Detailed results for radiative heat power simulation.

## Data Availability

The data presented in this study are available on request from the corresponding author.

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
