# Peer review of "Radiative Thermal Effects in Large Scale Additive Manufacturing of Polymers: Numerical and Experimental Investigations"

_materials, 2022, doi:10.3390/ma15031052_

Round 1
Reviewer 1 Report
Dear Authors,
I find Your work very interesting. However, there are some things that can be improved. Namely, the first chapter is hard to read and it should be broken into several passages.
In the Chapter 3, there are equations which are not explained in details, and thus are hard to understand (eq. 1 and 2). I'm not sure what exactly they're referring to.
Similar situation is with the equation 3 and 4. It's hard to visualize the reference of the equations without the figure to go with them. Chapter 4 is also very hard to follow.
Regards!
Author Response
Dear Reviewers,
Thank you for your very good comments and notes, which helped us to improve our article. We appreciate the feedback and have implemented your suggestions. Below, you can find details regarding the changes.
Best regards,
The Authors
Reviewer 1:
Dear Authors,
I find Your work very interesting. However, there are some things that can be improved. Namely, the first chapter is hard to read and it should be broken into several passages.
The introduction has been modified to be easier to read. The chapters have been set more precisely.
In the Chapter 3, there are equations which are not explained in details, and thus are hard to understand (eq. 1 and 2). I'm not sure what exactly they're referring to.
All terms of the equations have been clarified in fig 3 and fig 4.
Similar situation is with the equation 3 and 4. It's hard to visualize the reference of the equations without the figure to go with them.
All terms of the equations have been clarified in fig 3 and fig 4.
Chapter 4 is also very hard to follow.
Chapter 4 has been modified to be easier to read.
Regards!
Thank you for your comments!
Reviewer 2 Report
In this paper, the authors proposed a numerical model coupled with experimental data for the investigations of the LSAM process, which helped to understand some different thermal exchanges. Overall, this work is meaningful and practical for Large Scale Additive Manufacturing. Yet, some of the data need further clarification. Therefore, I recommend that several minor revisions of this paper are needed before publication.
Detailed comments listed below:
- What is the definition of the normalized time in Figure 6, Figure 8 and Figure 9?
- The authors mentioned that two types of measurements are performed. However, only the comparison of simulation and experiment by the radiative heat sensor was shown. The heat variation by IR camera was not mentioned.
- Different from Figure 6, there is a larger gap between simulation and experimental data in Figure 8. What is the reason for the discrepancy?
- The references are very few.
Author Response
Dear Reviewers,
Thank you for your very good comments and notes, which helped us to improve our article. We appreciate the feedback and have implemented your suggestions. Below, you can find details regarding the changes.
Best regards,
The Authors
Reviewer 2:
In this paper, the authors proposed a numerical model coupled with experimental data for the investigations of the LSAM process, which helped to understand some different thermal exchanges. Overall, this work is meaningful and practical for Large Scale Additive Manufacturing. Yet, some of the data need further clarification. Therefore, I recommend that several minor revisions of this paper are needed before publication.
Detailed comments listed below:
- What is the definition of the normalized time in Figure 6, Figure 8 and Figure 9?
Normalized time describes the elapsed time since starting the measurement divided by the measurement's total time. This has been clarified in the article (line 194).
- The authors mentioned that two types of measurements are performed. However, only the comparison of simulation and experiment by the radiative heat sensor was shown. The heat variation by IR camera was not mentioned.
The two measurements are:
1) Without extrusion of polymer for nozzle calibration
2) With extrusion of polymer for nozzle effect on already printed part. The IR camera measurement are used has input in the second numerical simulation of the radiative heat flux received by the IR sensor.
- Different from Figure 6, there is a larger gap between simulation and experimental data in Figure 8. What is the reason for the discrepancy?
Firstly, in fig 8 the speed of the nozzle is higher than in fig 6. The number of points per second in the experimental data recording is lower compared to the number of points of the numerical results.
Secondly, in fig 6, there is only the nozzle in the examined system (there is no extruded polymer). The experiment in fig. 6 are done to calibrate the numerical model of the extruder. That can explain the bigger difference in fig 8 when compared to fig 6.
- The references are very few.
Three more references have been added.
Le, A.D.; Cosson, B.; Asséko, A.C.A. Simulation of large-scale additive manufacturing303process with a single-phase level set method: a process parameters study.113, 3343–3360.304doi:10.1007/s00170-021-06703-5
Lepoivre, A.; Boyard, N.; Levy, A.; Sobotka, V. Heat Transfer and Adhesion Study for the346FFF Additive Manufacturing Process.47, 948–955. doi:10.1016/j.promfg.2020.04.291
Compton, B.G.; Post, B.K.; Duty, C.E.; Love, L.; Kunc, V. Thermal analysis of addi-348tive manufacturing of large-scale thermoplastic polymer composites.17, 77–86.doi:34910.1016/j.addma.2017.07.006
Thank you for your comments!
Reviewer 3 Report
The current manuscript studies the effect of thermal radiation in large-scale additive manufacturing (LSAM) of polymers. The title is attractive and some interesting results are presented. However, there are some issues that should be considered as follows:
- There should hook-up sentence in the abstract to show the importance and applications of LSAM.
- A critical review should be added to the introduction section including the latest research effort in this criteria.
- The introduction section should be ended with a paragraph that includes the problem statement and the main outline recommendations and experimental work of the current study. Other data related to the current work, starting from line #53, including Fig.1, should be moved to the experimental setup and methods sections.
- Fig, 2 a,b should be combined in one graph to avoid data redundancy. Also, the caption of Fig.2 is too short and it should contain more details.
- line #91, please revise the dimension of the build plate, it might be 1900 by 600 mm.
- More technical data about the radiative heat sensor and the IR camera should be added.
- Figures 3 and 4 should be combined, please avoid adding figures with repeated data.
- please revise the order of figures numbering, Fig.7 is stated before Fig.6 ?!
- There is a shortage of references in the results and discussion section that should justify the presented results and analysis.
- There should be an overlap between Fig.6 a and b in one plot to easily compare the data presented.
- The size of Fig.9 should be enlarged.
- The conclusion section should be in bullet points style to be easily tracked by the reader. This section should present the novelty, main contributions and results of the current work, in addition to the directly related applications.
- A grammatical and spelling check is required.
Author Response
Dear Reviewers,
Thank you for your very good comments and notes, which helped us to improve our article. We appreciate the feedback and have implemented your suggestions. Below, you can find details regarding the changes.
Best regards,
The Authors
Reviewer 3:
The current manuscript studies the effect of thermal radiation in large-scale additive manufacturing (LSAM) of polymers. The title is attractive and some interesting results are presented. However, there are some issues that should be considered as follows:
- There should hook-up sentence in the abstract to show the importance and applications of LSAM.
A hook-up sentence has been added to the abstract (By producing large components without geometrical constraints quickly and economically, LSAM processes have the capability to revolutionize many industries.).
- A critical review should be added to the introduction section including the latest research effort in this criteria.
Three more references have been added.
Le, A.D.; Cosson, B.; Asséko, A.C.A. Simulation of large-scale additive manufacturing303process with a single-phase level set method: a process parameters study.113, 3343–3360.304doi:10.1007/s00170-021-06703-5
Lepoivre, A.; Boyard, N.; Levy, A.; Sobotka, V. Heat Transfer and Adhesion Study for the346FFF Additive Manufacturing Process.47, 948–955. doi:10.1016/j.promfg.2020.04.291
Compton, B.G.; Post, B.K.; Duty, C.E.; Love, L.; Kunc, V. Thermal analysis of addi-348tive manufacturing of large-scale thermoplastic polymer composites.17, 77–86.doi:34910.1016/j.addma.2017.07.006
- The introduction section should be ended with a paragraph that includes the problem statement and the main outline recommendations and experimental work of the current study. Other data related to the current work, starting from line #53, including Fig.1, should be moved to the experimental setup and methods sections.
The divider between chapter 1 and chapter 2 has been reevaluated and we agree on the suggestion. Furthermore, the ending sentences of chapter 1 have been rewritten to better highlight the problem statement and the necessary investigations to address said problem.
- Fig, 2 a,b should be combined in one graph to avoid data redundancy. Also, the caption of Fig.2 is too short and it should contain more details.
We prefer to keep figures 2a and 2b separately, since 2a is the calibration and 2b is the measurement. Therefore, the setup changes noticeably between 2a and 2b, justifying the use of two figures.
The caption has been changed.
- line #91, please revise the dimension of the build plate, it might be 1900 by 600 mm.
You are correct, thank you! The dimensions have been revised.
- More technical data about the radiative heat sensor and the IR camera should be added.
Further information about the sensors has been added.
- Figures 3 and 4 should be combined, please avoid adding figures with repeated data.
Geometrical points and vectors have been added in the figure 3 and 4 in order to explain equations 1 to 4. Therefore, we prefer to keep them as separate figures.
- please revise the order of figures numbering, Fig.7 is stated before Fig.6 ?!
The order of figures has been corrected.
- There is a shortage of references in the results and discussion section that should justify the presented results and analysis.
Three more references have been added.
Le, A.D.; Cosson, B.; Asséko, A.C.A. Simulation of large-scale additive manufacturing303process with a single-phase level set method: a process parameters study.113, 3343–3360.304doi:10.1007/s00170-021-06703-5
Lepoivre, A.; Boyard, N.; Levy, A.; Sobotka, V. Heat Transfer and Adhesion Study for the346FFF Additive Manufacturing Process.47, 948–955. doi:10.1016/j.promfg.2020.04.291
Compton, B.G.; Post, B.K.; Duty, C.E.; Love, L.; Kunc, V. Thermal analysis of addi-348tive manufacturing of large-scale thermoplastic polymer composites.17, 77–86.doi:34910.1016/j.addma.2017.07.006
- There should be an overlap between Fig.6 a and b in one plot to easily compare the data presented.
The corresponding figure has been updated according to the recommendation.
- The size of Fig.9 should be enlarged.
The size of figure 9 has been enlarged.
- The conclusion section should be in bullet points style to be easily tracked by the reader. This section should present the novelty, main contributions and results of the current work, in addition to the directly related applications.
The conclusion section has been updated according to the suggestion.
- A grammatical and spelling check is required.
Grammar and spelling have been checked.
Thank you for your comments!
Round 2
Reviewer 1 Report
Dear authors,
thank You for the revisions.
Best regards.
Reviewer 3 Report
The revised manuscript is significantly improved and the review comments are well-considered.